# Fitness advantage of sequential metabolic strategies emerges from community interactions in strongly fluctuating environments

Zihan Wang[1,2], Yu Fu[3], Akshit Goyal 🄳[4]*, Sergei Maslov 🄳[1,2,5]*

**1** Department of Physics, The Grainger College of Engineering, University of Illinois Urbana-Champaign, Urbana, Illinois, United States of America, **2** Carl R. Woese Institute for Genomic Biology, University of Illinois Urbana-Champaign, Urbana, Illinois, United States of America, **3** Department of Physics, Yale University, New Haven, Connecticut, United States of America, **4** International Centre for Theoretical Sciences, Tata Institute of Fundamental Research, Bengaluru, India, **5** Department of Bioengineering, The Grainger College of Engineering, University of Illinois Urbana-Champaign, Urbana, Illinois, United States of America

\* maslov@illinois.edu (SM); akshitg@icts.res.in (AG)

## Abstract

Microbes growing in fluctuating environments employ two key metabolic strategies: sequential (diauxic) utilization and co-utilization of nutrients. Most work has focused on understanding and comparing these strategies physiologically for the growth of single species, rather than ecologically for the assembly of complex natural communities. This is in part because of the lack of a framework for directly comparing the fitness of these strategies in an ecological context. Here, we present a new consumer-resource framework that incorporates dynamic proteome reallocation, and use it to compare the fitness of metabolic strategies during community assembly. We introduce two notions of fitness of a strategy in fluctuating environments: the time-averaged growth rate and the biomass-weighted prevalence of microbes using a given strategy. We find that sequential utilizers, although disadvantaged in pairwise competitions, gain a significant edge during community assembly — an advantage that becomes more pronounced with increasing community diversity and the size of the species pool from which they are assembled. Low diversity communities resemble pairwise competitions and are dominated by co-utilizers, whereas high diversity, mature communities (i.e., those assembled from a larger species pool) are dominated by the sequential utilizers. This shift is driven by two factors: the difference in lag times and the increased structural stability conferred by sequential strategies. Overall, our work provides several testable predictions about the co-occurrence patterns of microbes using different metabolic strategies.

**Data availability statement:** There are no data associated with this paper. All code is available as a GitHub repository at the following link: https://github.com/maslov-group/Ecol-adv-diaux.

**Funding:** The research of Z.W. was supported in part by grants from the NSF (DMS-2235451) and Simons Foundation (MPS-NITMB-00005320) to the NSF-Simons National Institute for Theory and Mathematics in Biology (NITMB), with S. M. as the grant recipient. A. G. was supported by the Ashok and Gita Vaish Junior Researcher Award, the DST-SERB Ramanujan Fellowship, and the DAE, Government of India, under project no. RTI4001. The funders had no role in study design, data collection and analysis, decision to publish, or preparation of the manuscript.

**Competing interests:** The authors have declared that no competing interests exist.

## Author summary

In nature, microbes often face "boom-and-bust" environments where nutrients arrive in sudden bursts and are then slowly depleted. To survive, species choose between two main metabolic strategies: consuming all available nutrients together (co-utilization) or utilizing them one-by-one (sequential utilization). Traditionally, studies have compared these strategies focusing on single species in isolation. We found, however, that the competitive advantage of a strategy depends significantly on its ecological surroundings. Using a new modeling framework, we show that sequential species are poor competitors in isolation or communities with a few species, due to the time lost switching between food sources. However, in a sufficiently diverse community, sequential species collectively have an overwhelming competitive edge. This occurs because different sequential species in a diverse community naturally assemble to be complementary to each other by partitioning which nutrient they most prefer. This collective behavior minimizes individual switching delays, allowing sequential utilizers to dominate communities assembled from highly-diverse pools of species. Furthermore, we show that communities enriched with sequential species exhibit are also resilient to environmental fluctuations in nutrient supply. Our findings help explain how diverse metabolic strategies coexist in nature.

## Introduction

Microorganisms use a variety of metabolic strategies to utilize resources in their environments. They typically either consume the available resources all at once (co-utilization) or one after another (sequential utilization) [1–3], as shown in Fig 1A-B. Both strategies have different effects on physiology, i.e., on the growth rate of a single species in isolation. Even in this context, the answer to which strategy is superior remains disputed. Sequential strategies have been theoretically shown to be growth-optimal for certain pairs of resources and co-utilization optimal for others, which depends on the position of the resources in the central metabolic network [4]. However, experimental studies [5–9] show that microbes often deviate from growth-optimality, by allocating part of their proteome to resources that are not currently utilized [10].

While understanding the advantages of using different metabolic strategies when species are grown in isolation is important, microbes in natural environments rarely grow alone. Instead, they live in complex multi-species ecosystems, where emergent factors that are distinct from growth rate alone may favor some strategies over others. These ecological effects have never been systematically studied, either experimentally or theoretically. Thus, studies of both strategies in an ecological context may shed light on how and why they may coexist in natural environments.

An intuitive understanding of how ecological contexts might change the competitive advantage of a metabolic strategy is as follows. When growing in isolation,

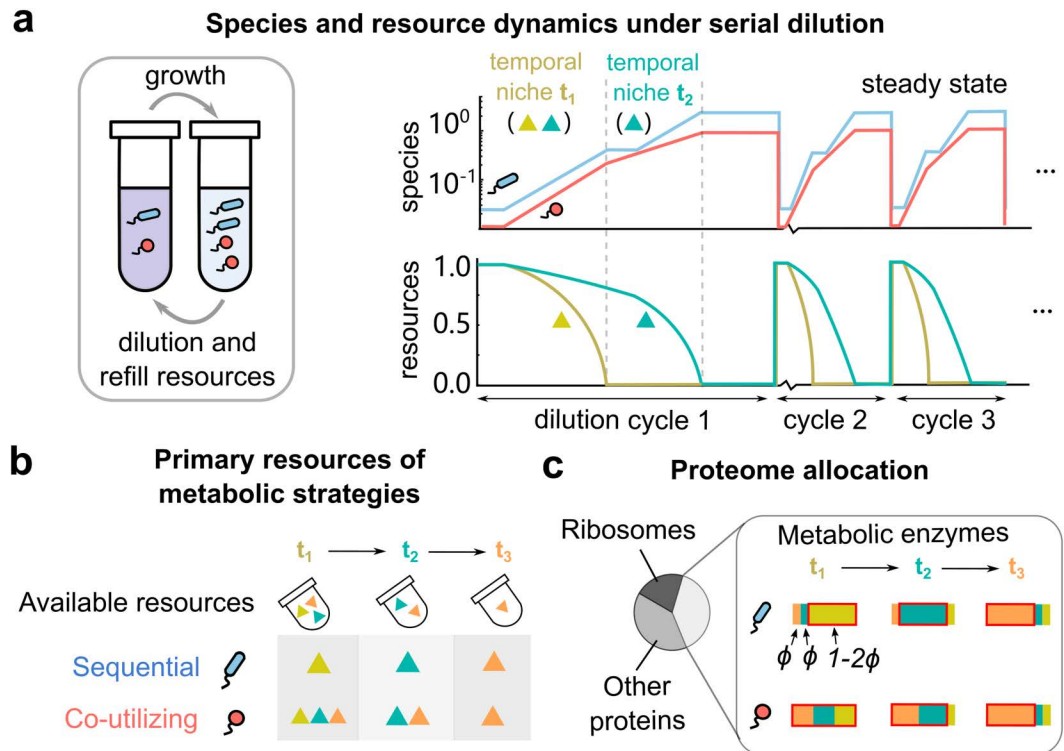

**Fig 1. Investigating sequential and co-utilizing metabolic strategies in boom and bust environments requires incorporating proteome allocation in consumer-resource models. (a)** Schematic illustration of species and resource dynamics in boom-and-bust (serial dilution) environments. In each growth-dilution cycle, resources are depleted in a specific temporal order, giving rise to "temporal niches" defined by which resources are currently available. Here, in the first niche lasting time $t_1$, both resources (gold and teal) are present, while in the second niche lasting time $t_2$, only one resource (teal) is present. After several growth-dilution cycles, both species (red and blue) reach a steady state where they grow by exactly the dilution factor $D$ in each cycle. After a resource is depleted, species using it experience a lag time depending on how they preallocate internal enzymes (Methods). **(b)** Primary resources of two major metabolic strategies: sequential (blue) and co-utilizing (red). Species using sequential strategies always consume one resource at a time, according to their idiosyncratically hardwired preference order (here, gold, then teal, then orange). Co-utilizing species use all available resources at the same time. **(c)** Illustration of proteome (re)-allocation in our model. We assume that each species has a proteome broadly divided into three sectors: ribosomes, metabolic enzymes, and other housekeeping proteins. Within the metabolic sector, all species allocate a fixed, small fraction $\phi$ to each secondary resource according to the metabolic strategy, and the rest of the metabolic sector (red box) is equally allocated to all primary resource(s).

species can always count on resources being present. In contrast, in communities, some resources may be depleted by other species before a species gets a chance to consume them [11]. This becomes only more likely as the number of species (community diversity) increases. Such an effect may lead to differences in the performance of a strategy between pairwise competitions and community contexts.

Fluctuating environments such as in serial dilution experiments — where resource availability strongly changes with time — present a natural scenario in which to compare sequential and co-utilizing strategies (Fig 1A). Indeed, in steady-state environments such as chemostats, resources reach low concentrations, where even sequential utilizers become co-utilizers [1,3]. Consequently, the temporal niche segregation that facilitates the success of sequential strategies cannot manifest in such systems where resource concentrations remain invariant over time.

Here, using simulations and theory, we quantify the advantages of metabolic strategies in complex communities using two metrics of fitness of microbes using a given strategy: the time-averaged growth rate, which serves as a mechanistic

indicator to explain the origins of competitive advantage, and the biomass-weighted prevalence, which reflects the final ecological outcome of the community assembly process. We find that sequential strategies have a systematic fitness advantage over co-utilizing strategies, which emerges in community contexts, getting stronger with increasing community complexity (number of species) and the size of the species pool from which communities are assembled. The size of the species pool ($N_{pool}$) effectively represents the "maturity" of the community assembly process, as it corresponds to the cumulative number of species that have attempted to invade and colonize the environment over time. Throughout this text, we refer to communities assembled from such larger species pools as "mature communities." In our simulations, low diversity, less mature communities are dominated by co-utilizers, while high diversity and more mature communities are dominated by sequential species. We then investigate the source of this behavior, and find two key contributors. They are (1) the decreased importance of lag time differences between sequential utilizers and co-utilizers in ecological contexts, and (2) increased resilience to resource fluctuations (structural stability [12–14]) of communities enriched in sequential utilizers. Our results have implications for how both sequential and co-utilization strategies coexist in nature.

## Results

### Modeling dynamic proteome allocation between primary and secondary resources

Microbes inhabit fluctuating environments and so they allocate their proteome dynamically: as conditions change, cells retune the metabolic sector to match the resources they rely on at that moment. Here, we develop and study a model of community assembly from a pool of microbial species using two broad metabolic strategies of resource utilization: sequential and co-utilization [4,10]. In the sequential (diauxic) strategy, most of the metabolic proteome is devoted to the most-preferred (highest-ranked) resource among those currently available. In this strategy, we refer to that top-ranked resource at a given time as a primary resource. In the co-utilization strategy, multiple resources are consumed simultaneously, and we refer to each currently consumed resource as a primary resource. For simplicity, we assume that co-utilizers allocate enzymes evenly across all their primary resources. For both strategies, we refer to resources that are not currently being consumed, including depleted ones, as secondary resources.

To quantify how different microbes deal with resource depletion, we introduce a secondary allocation fraction $\phi$ that represents the share of the metabolic proteome invested towards each secondary resource (Fig 1C). This equal per-resource investment creates a pool of metabolic enzymes that prepares an organism for possible shifts to utilizing other resources once they deplete those they are currently utilizing. This assumption is consistent with observations that microbes often allocate enzymes to resources they are not presently consuming [5,15–17]. Further, based on environmental context, this allocation can either be static or dynamic. Some species follow a "set-it-and-forget-it" strategy, maintaining small fixed allocations to certain transporters or enzymes regardless of immediate need [6,7]. In contrast, studies of different strains of *Saccharomyces cerevisiae* show that they actively modify their secondary proteome allocation in anticipation of upcoming resource depletion [15]. Consequently, in our model we treat $\phi$ as an effective parameter that summarizes the average level of preparedness for shifts in resource availability. We make this assumption in favor of simplicity to avoid having to model the complex details of the dynamics of $\phi$ [18]. Physiologically, $\phi$ represents the basal proteomic investment in enzymes for resources that are not currently serving as a primary growth source, whether they are suppressed by regulatory hierarchies [10,19] or are physically depleted from the environment. This investment is fundamentally limited by the total metabolic proteome capacity, requiring $n_R\phi \leq 1$ to remain physically feasible.

The parameter $\phi$ plays a central role in our model because it governs lag times after resource depletion. Larger $\phi$ shortens the time needed to initiate growth in this new environment by seeding the required enzyme pools in advance, but it also reduces the instantaneous growth rate on currently used resources by diverting part of the metabolic budget [5,8,9]. When a primary resource is depleted, the allocation profile is rebalanced: hierarchical species shift most of their proteome to the next resource in their priority list (which then becomes a primary resource), whereas co-utilizers redistribute across

the remaining primary resources. The extent of proteome reallocation directly determines the lag time: larger redistributions prolong the lag, while smaller adjustments make it shorter.

Beyond internal allocation within the metabolic sector of the proteome, in our model, we also account for dynamic reallocation between the ribosomal and metabolic sectors. Following the allocation rule established by Ref. [20,21], both sectors adapt to the instantaneous growth rate afforded by the currently utilized resources, ensuring that the proteome remains balanced as environmental conditions change. We assign each species a set of enzyme coefficients ($\xi_{\alpha k}$) for each resource; the instantaneous growth rate is then determined by these coefficients weighted by the proteome fraction allocated to the corresponding enzymes (Eq. 1, Methods). Importantly, we assume that for any species, enzyme coefficients in different single-resource environments are selected independently, so there is no trade-off between growth rates across environments. In this study, we systematically span a range of realistic values of the secondary allocation fraction $\phi$ — from 0% to 10% — and directly compare the relative advantage of either using the hierarchical or co-utilization strategy.

### Emergence of ecological advantage of sequential strategies

One way to compare different metabolic strategies is to measure the overall, or time-averaged growth rate of species utilizing them. This is straightforward when species are growing alone and can be done simply by taking logarithm of the factor by which a species grows over a single boom-bust cycle (at steady state, this is always the dilution factor $D$) and dividing it by the time it takes to deplete all resources: $\langle g \rangle_t = \frac{1}{T_{\text{dep}}} \log \frac{N(T_{\text{dep}})}{N(0)}$ (Fig 2A). This metric closely resembles the notion of average fitness of a species, since it reflects a time-averaged growth rate. We measured the time-averaged growth rate for single species utilizing two specific strategies: co-utilization and the "top-smart" sequential strategy, where a species first utilizes the resource on which it grows fastest, then uses all other resources in an idiosyncratic order uncorrelated with growth rate. We previously identified this as the dominant sequential strategy in boom-and-bust environments under random community assembly [11]. While we focus on this strategy in the main text, we find that our results are qualitatively robust to relaxing this assumption. In Fig A in S1 Text we show that sequential utilizers with random preference orders also show the same qualitative trends, albeit with minor quantitative differences.

We found that in a single species (monoculture) context, co-utilizers have significantly higher time-averaged growth rates than sequential species (Fig 2C) over the entire range of the secondary allocation fraction $\phi$ tested. This observation can be attributed to the much lower lag times of co-utilizers compared with sequential species. Indeed, the difference in time-averaged growth rates between strategies almost disappears if we unrealistically eliminate the lag times of both strategies (Fig B in S1 Text). This suggests that in single-species contexts, co-utilizers are more fit than sequential species.

However, in a community context, all surviving species must have the same time-averaged growth rate in order to coexist. Thus if co-utilizers and sequential species coexist, it will be impossible to compare them on the basis of their time-averaged growth rates. To disentangle how such community coexistence—rather than individual physiology—modifies a strategy's fitness, we assembled "pure" communities comprising species utilizing only one of the two strategies. To maximally distinguish community contexts from single-species contexts, we assembled communities of maximal diversity, where the number of species equaled the number of resources. Surprisingly, in such community contexts, the time-averaged growth rates of communities comprising co-utilizers and sequential species were comparable (Fig 2D). At low values of $\phi$, sequential species had a slight advantage in time-averaged growth rate, while at high values of $\phi$, co-utilizers grew faster on average. This drastic reduction in the growth rate disparity between both strategies in ecological contexts can be explained as follows. As communities become more diverse, coexisting sequential species tend to be complementary to each other, i.e., they prefer different resources as their top choice [11]. Because of this complementarity, once species deplete their top choice resource, they often have no resources left to switch to since they have been depleted by others. Hence, the lag disadvantage of sequential species becomes essentially irrelevant in diverse community contexts.

 

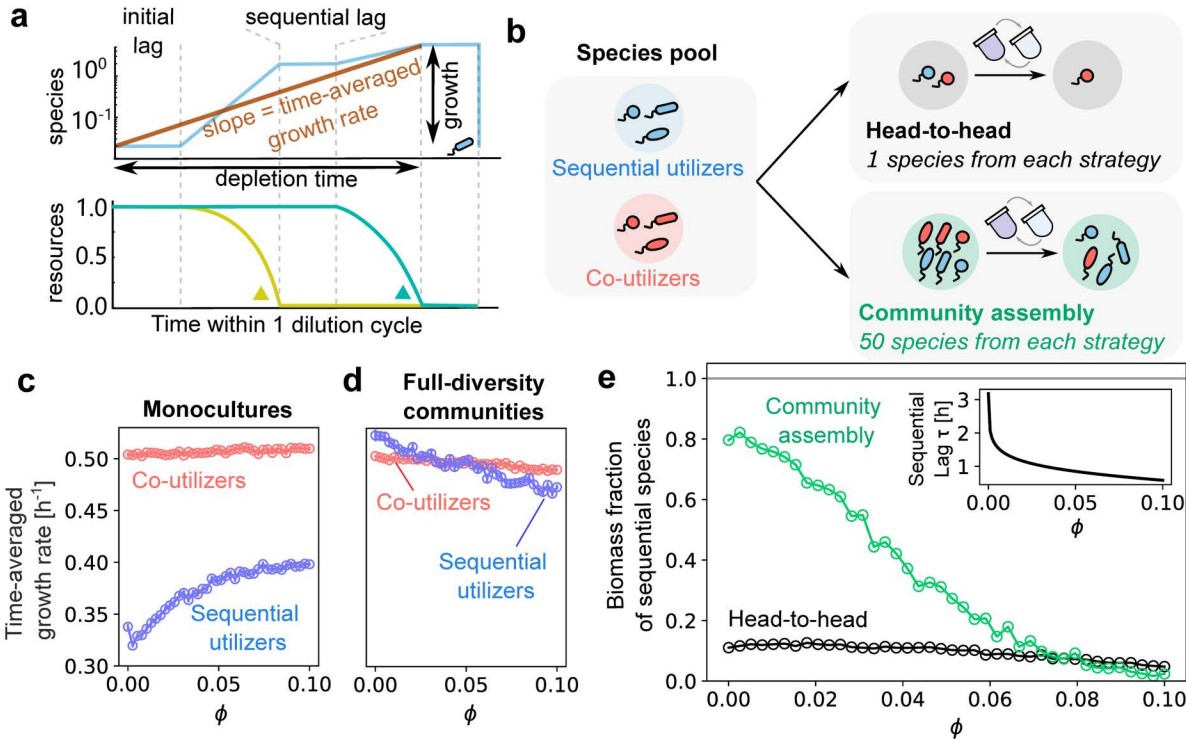

**Fig 2. Comparison of co-utilization and sequential strategies in single-species and community contexts. (a)** Schematic of time-averaged growth rate measurement. At steady state, the time-averaged growth rate is inversely proportional to the time it takes to deplete all resources. Top: species growth dynamics, including initial and sequential lags. Bottom: corresponding resource depletion over time. **(b)** Schematic showing our simulations in two different scenarios: one where we compete 1 sequential and 1 co-utilizing species head-to-head (gray), and the other where we assemble a complex community seeded with 50 species from each strategy (green). **(c)-(d)** Time-averaged growth rates of co-utilizers and sequential utilizers in monoculture and in "pure" communities of the same strategy with maximum diversity as a function of secondary allocation fraction $\phi$. In monoculture **(c)**, co-utilizers consistently exhibit higher growth rates due to lower lag times. In full-diversity communities **(d)**, sequential utilizers' time-averaged growth rates become comparable to those of co-utilizers, and even exceed the latter at low $\phi$. **(e)** Fraction of surviving sequential utilizers in each of the two scenarios: head-to-head (black) and in complex communities (green), plotted as a function of the secondary allocation fraction $\phi$ common to all species in the pool. Sequential utilizers have a distinct ecological effect in complex communities: the green curve is typically above the black curve. Inset: average lag time $\tau$ during the first resource switch of sequential utilizers, as a function of $\phi$.

Indeed, it is short depletion time—not species' biomass—that is ultimately selected for in these environments. Consequently, in community contexts, sequential species can often be as fit, if not fitter, as co-utilizers.

Having established this mechanistic intuition, we move to the definitive standard of ecological success: the biomass-weighted prevalence of a strategy in naturally assembled mixed communities. To measure this, we assembled several multi-species communities, each starting from a large species pool with 50 species utilizing a sequential strategy and 50 species which were co-utilizers (Methods; Fig 2B). We serially diluted these communities until they reached steady state. We measured the prevalence of sequential species as the biomass-weighted fraction of them among all survivors in assembled communities. We found that just like we observed in Fig 2D, in assembled communities sequential utilizers had much higher prevalence than co-utilizers (reaching as high as 85%; Fig 2E). Their prevalence gradually decreased with increasing $\phi$, reaching parity (50%) at $\phi \approx 0.03$. This confirms and amplifies the ecological effect we observed in time-averaged growth rates in Fig 2C-D. Even the $\phi$ values at which both strategies reach parity are comparable. Note that in contrast with the scenario in Fig 2D, which was somewhat artificially constructed, the advantage of sequential species in Fig 2E emerges naturally through the process of community assembly. This advantage persists in the absence

of lag times, albeit it gets weaker and can be explained in terms of the statistics of growth rate distributions (Fig B in S1 Text). While we assume that our species pool had an equal proportion of both strategies to ensure a fair comparison, we also observed that sequential species were consistently enriched and often dominated even when starting as a smaller fraction of the species pool (Fig C in S1 Text). This reinforces the robustness of their ecological advantage. Together, these results point to the emergence of a distinct and significant advantage for sequential species in ecological contexts of species-rich communities.

## Sequential strategies promote diversity through increased structural stability

To better understand the forces guiding the assembly of complex communities of species using different metabolic strategies, we investigated the composition of communities with varying levels of species diversity or complexity. Because of competitive exclusion, the final species diversity ranged between 1 and the number of resources $n_R = 4$, depending on the randomly generated species pool. Interestingly, the fraction of sequential species was strongly correlated with this final species diversity (Fig 3A). That is, across all values of $\phi$ tested, sequential species dominated in high-diversity assembled communities, while co-utilizers were prevalent in low-diversity communities (Fig 3A). Fig 3B shows a detailed breakdown of the community diversity at different values of $\phi$. As $\phi$ increases, the most common community diversity shifts from maximally diverse (4 surviving species on 4 resources) to minimally diverse (1 surviving species). Further, at each value

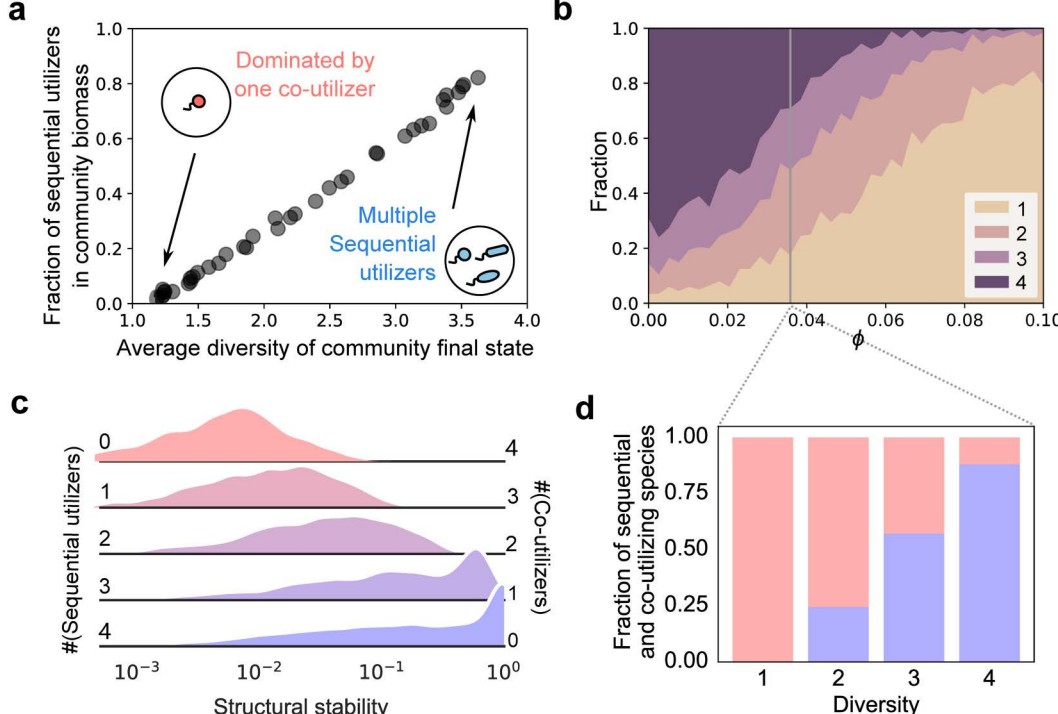

**Fig 3. Sequential strategies promote diversity through increased structural stability. (a)** Fraction of surviving sequential species in biomass as a function of the average diversity of the assembled communities (# of surviving species) for simulated communities across all $\phi$ values tested. As community complexity increases, sequential species become more dominant in communities. **(b)** Stackplot showing how the diversity of assembled communities are stratified across $\phi$. At low $\phi$ the majority of communities have full diversity, while at high $\phi$ most of them are dominated by 1 species. **(c)** Distributions of normalized structural stability (Methods) for niche-packed communities ($n_S = n_R$ species). Increasing the number of sequential utilizers, from 0 to 4 systematically increases structural stability. **(d)** Stacked bar plot showing how sequential (red) and co-utilizing species (blue) are stratified across communities with different diversities at a fixed value of $\phi \approx 0.036$. Here too, more complex communities are increasingly dominated by sequential species, while less complex communities are dominated by co-utilizing ones.

of $\phi$, strategies were segregated by community diversity. That is, single-species communities almost entirely contained co-utilizers ($\approx$100%) while maximally diverse communities with 4 species were dominated by sequential utilizers (Fig D in S1 Text). This segregation of strategies by diversity was observed even when $\phi \approx 0.036$ (Fig 3D), when sequential and co-utilizing species were on average equally prevalent in communities (Fig 2E).

A plausible explanation for the stratification of strategies by community richness/diversity (number of species) lies in the systematic difference in their structural stability [12], i.e., the range of resource supply conditions ($R_k$) within which a given set of species can stably coexist (Fig E in S1 Text). A community with greater structural stability has a larger domain of feasibility, which implies that it can withstand harsher environmental fluctuations without losing species. Mechanistically, communities containing species with niche overlap will have greater structural stability (Methods). In our case, niche overlap is determined primarily by differences in consumption ratios for co-utilizers and complementarity in resource preferences for sequential utilizers [11]. As an example, a community containing sequential species with complementary top choice resources will have lowest niche overlap and thus high structural stability. While structural stability can be determined numerically using Monte Carlo techniques [22], it is generally computationally expensive. To circumvent this computational load, we developed a simple and fast linear algebra approach to calculate the structural stability for any niche-packed community in our model (Methods). By applying our method to niche-packed communities composed of different number of sequential species (from 0 to $n_R = 4$ at $\phi \approx 0.036$), we found that structural stability progressively increased with increasing number of sequential species, becoming the largest in communities composed entirely of sequential species (Fig 3C, bottom). As before, these results are qualitatively robust to variation in lag times (Fig B and F in S1 Text). This suggests that for a given set of resource supply concentrations, sequential species would tend to be over-represented in diverse niche-packed communities.

## Discussion

In this manuscript, we introduced and measured two different notions of fitness of metabolic strategies in time-varying environments. These notions were (a) the time-averaged growth rate of microbial species using each strategy, and (b) the prevalence of species using each strategy in stochastically assembled microbial communities. Our central result is that sequential strategies become progressively fitter in ecological contexts over co-utilizing strategies using both notions of fitness. The observed fitness difference becomes especially pronounced with increasing community diversity as measured by species richness, as well as the size of the species pool from which communities are assembled.

It is important to understand how the different key parameters in our model affect the strength of this central result. The most explicit of these parameters is the secondary allocation fraction $\phi$ which represents the fraction of the proteome that species pre-allocate to resources that they do not consume in a particular environment. The advantage of sequential utilizers persisted for a reasonably broad range of $\phi$ (Fig 2C-E). Specifically, low values of $\phi$ increasingly favored sequential species, while higher values favored co-utilizers. The overwhelming advantage of co-utilizers in monoculture contexts comes from their significantly lower lag times. In community contexts, complementary sequential utilizers — with distinct top-choice resources — can assemble to form communities where each species essentially depletes all resources together, and species barely derive any growth from resources other than their top choice. Such collective growth makes the lag disadvantage of sequential species immaterial, contributing to their increased time-averaged growth rate (fitness) in ecological contexts (Fig 2D).

In addition to $\phi$, the lag times depend on the prefactor $\tau_0$ (Methods), which sets an overall scale for lag times. With no lags ($\tau_0 = 0$, we no longer see a stark difference between time-averaged growth rates in monoculture versus ecological contexts (Fig B in S1 Text). Sequential species still retain a "biomass advantage" in communities as measured by prevalence (Fig B in S1 Text, panel A green). The advantage in this case can be explained by a rather different effect: namely the statistics of growth rate distributions for species utilizing either strategy (Fig B in S1 Text, panels B-C). The odds of a sequential species to win in a head-to-head competition with a co-utilizing species depend on which strategy's growth

rate distribution has the larger median, which at $\phi \approx 0.06$ is the co-utilizing strategy (Fig B in S1 Text, panel C). However, during community assembly from a large species pool, the level of competition is much greater since only $n_R = 4$ species can survive out of the pool of $N_{pool} = 100$ introduced species. In this scenario, it is not the medians, but the extreme value of the growth rates that matter. The extremes are determined by the variance of the growth rate distributions, which is larger for the sequential strategy. This is because, unlike co-utilizing strategies, sequential species do not average over $n_R$ growth rates, one for each resource, to determine their growth rate in any environment. Hence, for the same $\phi \approx 0.06$, it is the sequential strategy, not the co-utilizing strategy, that does better during community assembly (Fig B in S1 Text, panel B).

Thus, not just lag times, but the width of the growth rate distributions also contribute to the success of metabolic strategies. In Fig G in S1 Text, we systematically investigate the effect of changing this width $\sigma$, finding that lower values of $\sigma$ lead to a lower fitness of sequential strategies as measured by their prevalence in assembled communities (Fig G in S1 Text, panel A). This is consistent with above observation that the success of sequential strategies in communities is determined by the extreme value distributions of growth rates, whose median depends positively on $\sigma$. Furthermore, our findings are robust to, and even strengthened by, the inclusion of physiological trade-offs in enzyme efficiencies. Indeed, we find that imposing tradeoffs in enzyme efficiencies actually enhances the dominance of sequential strategies (Fig H in S1 Text). The reason behind this enhancement is that imposing such tradeoffs serves to increase the width of the growth rate distribution $\sigma$, which as we explained promotes sequential utilizers.

The degree to which the time-varying environments can be considered boom-and-bust environments depends on the dilution factor $D$, with increasing $D$ leading to stronger busts. We found that changing $D$ had a consistent but small effect of changing the fitness of sequential strategies in community contexts, across a wide range of experimentally relevant values ($D = 10$–$10^4$, Fig I in S1 Text). In particular, lag times become more important for lower values of $D$, since at steady-state, the lag time is fixed and independent of $D$, while the duration of the growth phase increases in proportion to $\log D$. As a result, as $D$ decreases, so does the fitness of sequential utilizers in community contexts as measured by biomass prevalence (Fig I in S1 Text, panel A green) and time-averaged growth rates (Fig J in S1 Text).

Our other key result is that communities with sequential species are also significantly more structurally stable, and thus able to withstand greater fluctuations in resource supply ratios. To understand why this is the case, consider the following argument. Sequential species use only one resource at a time. Importantly, some sequential species may never get to utilize an available resource before it gets depleted by others in the community. For example, if the most preferred resource by a species gets depleted last by the community, this species will never get to use another resource, even though it might be capable of using all resources. In contrast, co-utilizing species always consume all available resources regardless of their depletion order. As a consequence, changing the concentration of any resource will affect all co-utilizers but only a fraction of sequential species. In more mathematical terms, the matrix $M$ encoding which species consume which resources in the steady state cycle (Methods) will be sparser for sequential communities than co-utilizing ones and hence be more structurally stable. With increasing number of resources, the $M$ matrix would get larger and thus we expect the systematic difference between sequential and co-utilizing species to increase. We explore structural stability more systematically, with its dependence on model parameters, in the supplementary materials (Fig F, G, and I in S1 Text).

While our simulations utilized $n_R = 4$ resources, our results generalize to communities with higher diversity. Mathematically, the normalized structural stability $s_R$ for sequential communities remains largely independent of $n_R$, and the reduction in niche overlap becomes even more pronounced as the potential for temporal partitioning increases. However, the dominance of sequential species in such systems depends on the size of the species pool ($N_{pool}$). Since sequential success is driven by extreme value statistics, increasing $n_R$ requires a larger pool to ensure the presence of niche-complementary specialists for every resource. Thus, $n_R = 4$ serves as a representative case that captures the emergence of community-level advantages without requiring computationally prohibitive species pool sizes.

It is also important to distinguish the ecological advantages identified here from those in steady-state environments, such as chemostats. In our "boom-and-bust" framework, the success of sequential strategies relies on temporal niche

partitioning, which allows communities to mitigate the physiological cost of switching lags. In a steady-state system, resource concentrations are invariant over time. While sequential utilizers might still persist as specialists in such systems, the specific mechanism of temporal complementarity and its associated boost to structural stability are intrinsically tied to environmental fluctuations. Exploring how these strategies fare in constant resource regimes would require accounting for different physiological and ecological mechanisms, which is beyond the scope of this work.

Finally, we discuss the potential impact of environmental complexity beyond our simplified model. In this work, we assumed a well-mixed environment where species compete only for externally supplied resources, neglecting spatial structure and cross-feeding for simplicity. While these factors are prevalent in natural communities [23–26], we argue that the results presented here are qualitatively robust to the inclusion of spatial heterogeneity or cross-feeding interactions. Mechanistically, cross-feeding would tend to create resources that species can only consume later during a cycle, and would thus only contribute to later temporal niches. Thus, the contribution of these cross-feeding niches to overall growth of species would be relatively low, resulting in only a small quantitative change to our results. Similarly, while detailed spatial modeling is beyond the scope of this study, Including spatial heterogeneity would likely introduce "spatial niches" that operate in addition to the temporal niches described here.

## Testable predictions

Our results might potentially explain the diversity of metabolic strategies observed in nature, specifically why sequential (diauxic) and co-utilizing strategies can coexist without one systematically outcompeting the other across all environments. In our model, both strategies stratify in communities with different diversity, or species richness, relative to the number of resources (Fig 3A, D). This leads to the following testable prediction: niche-packed communities ($n_S/n_R \approx 1$) would tend to be richer in sequential species, while poorly packed ($n_S/n_R \ll 1$) communities would be dominated by co-utilizers. Future experimental or observational work could potentially test this prediction and help explain the diversity of observed metabolic strategies.

Another observation of our work is that the fraction of sequential species in communities increases with the species pool size (Fig K in S1 Text). The species pool size represents the "maturity" of a community, i.e., the cumulative number of invasion attempts by other species into a community. While co-utilizers often persist in low-diversity communities, the fitness advantage of sequential strategies is systematically amplified with increasing community diversity. We predict that more mature communities in nature will tend to be enriched in sequential species.

## Methods

### Model of microbial proteome allocation

To incorporate proteome allocation into our consumer-resource model [23,27–31] in boom-and-bust environments, we extended previous proteome allocation schemes to include pre-allocation of metabolic enzymes. That is, just as in previous models, we assumed that the proteome of each individual in a species $\alpha$ was composed of metabolic enzymes ($E_\alpha^{(m)}$), ribosomes ($E_\alpha^{(r)}$), and housekeeping proteins ($E_\alpha^{(h)}$) [20].

We perform proteome allocation in two steps. First, we assume that the allocations to the housekeeping sectors are fixed, and the rest of the proteome is dynamically allocated to metabolic and ribosomal sectors. By rescaling, we can write $E_\alpha^{(m)} + E_\alpha^{(r)} = 1$.

Secondly, we subdivide the metabolic sector into distinct enzymes, where we assume each enzyme is specific to one resource. While multi-substrate (promiscuous) enzymes exist in nature, our model assumes one-to-one mapping as a standard approximation. This assumption allows us to focus on the proteomic costs associated with shifting between distinct metabolic pathways, which is the primary driver of the lag dynamics in our model. Consider a species $\alpha$ growing

in a particular resource environment. To grow on these resources, the species allocates a certain fraction of metabolic enzymes $x_{\alpha k}$ to resource $k$. Note that in our model, species allocate enzymes even towards secondary resources. Based on proteome allocation models [20,32], the growth rate $g_\alpha$ is determined by the balance between the protein synthesis flux ($J_{\text{trans}}$) and the metabolic uptake flux ($J_{\text{met}}$). The translation flux is proportional to the ribosomal sector $E_\alpha^{(r)}$: $J_{\text{trans}} = g_C E_\alpha^{(r)}$, where $g_C$ is the translation efficiency. Similarly, the metabolic flux depends on the allocation of metabolic enzymes $E_\alpha^{(m)}$ and their efficiencies on primary resources: $J_{\text{met}} = \left( \sum_{k \in S_{\text{primary}}} \xi_{\alpha k} x_{\alpha k} \right) E_\alpha^{(m)}$, where $\xi_{\alpha k}$ stands for a species-specific coefficient representing the contribution of given resource $k$ towards growth per unit enzyme fraction. $S_{\text{primary}}$ is the set of primary resources.

At steady state, the cell optimizes its proteome by matching these two fluxes ($g_\alpha = J_{\text{trans}} = J_{\text{met}}$). By imposing this flux-matching condition and using the normalization constraint $E_\alpha^{(m)} + E_\alpha^{(r)} = 1$, we can solve for the overall growth rate $g_\alpha$ to obtain:

$$g_\alpha = \left( \frac{1}{\sum_{k \in S_{\text{primary}}} \xi_{\alpha k} x_{\alpha k}} + \frac{1}{g_C} \right)^{-1}.$$

(1)

We then consider how species allocate $x_{\alpha k}$ towards different metabolic enzymes. With $n_R$ resources in total, we assume that a $\phi$ fraction of the metabolic sector is allocated to each secondary resource, and the rest is equally allocated to all the primary resources:

$$x_{\alpha k} = \begin{cases} \phi + \frac{(1 - n_R \phi)}{|S_{\text{primary}}|} & \text{if } k \in S_{\text{primary}} \\ \phi & \text{if } k \notin S_{\text{primary}} \end{cases}$$

(2)

$S_{\text{primary}}$ for sequential utilizers is the single most preferred available resource. For co-utilizers, it is the set of all available resources.

To understand how $\phi$ affects the balance between sequential and co-utilizing strategies in head-to-head competition, suppose there are two strains $c$ and $s$ sharing the same set of physiological parameters ($\xi_{ck} = \xi_{sk} = \xi_k$), but $c$ is a co-utilizing strain, while $s$ is a top-smart sequential strain (grows fastest on its most preferred resource). These strains are grown in a boom-and-bust environment with $n_R$ resources at the beginning of each cycle. When they compete against each other, our previous work shows that the competition occurs mainly during the first temporal niche, where all resources are present [4]. To compare the growth rates of the sequential and co-utilizing species, we only need to compare their $\sum_{k \in S_{\text{primary}}} \xi_{\alpha k} x_{\alpha k}$. The ratio $r_{sc}$ quantifies the advantage of the top-smart sequential strategy over the co-utilizing one. It is given by theh following equation when all substrates are present:

$$r_{sc} \equiv \frac{\max_k(\xi_k x_{sk})}{\sum_i^{n_R} \xi_k x_{ck}} = \frac{\max_k(\xi_k) \cdot \left( \phi + 1 - n_R \phi \right)}{\sum_k^{n_R} \xi_k / n_R}.$$

(3)

Note that for $\phi = 0$ the sequential strategy always has an advantage in the first temporal niche: $r_{sc} > 1$. Indeed, in this case $r_{sc}$ is given by the ratio of the maximum over the mean growth rate, which is always larger than 1. As $\phi$ increases, a larger re-allocation portion of metabolic enzymes starts to favor the co-utilization strategy over the sequential utilization one, so that at some value of $\phi$ the ratio would satisfy $r_{cs} < 1$. When lags are absent, this argument qualitatively explains the behavior of the fraction of surviving sequential utilizers in head-to-head competition shown in Fig A in S1 Text, panel A (black curve).

## Model of microbial community dynamics in boom-and-bust environments

We model microbial community dynamics in a boom-and-bust environment where $n_R$ substitutable resources are cyclically supplied at concentrations $R_k(0)$ ($k = 1, 2, \ldots n_R$). We attempt to assemble a community by adding species to this environment. Each species is assumed to be a generalist able to grow on each of these resources. We denote the abundance of species $\alpha$ as $N_\alpha(t)$, and the concentration of resource $k$ is represented by $R_k(t)$. To capture the metabolic strategy, we define a consumption matrix $c_{\alpha k}(t)$:

$$c_{\alpha k}(t) = \begin{cases} 1 & \text{if } k \in S_{\text{primary}}^{(\alpha)} \text{ at } t \\ 0 & \text{otherwise} \end{cases}$$

(4)

As a result, we can rewrite Eq. (2) as

$$x_{\alpha k}(t) = \phi + (1 - n_R \phi) \cdot \frac{c_{\alpha k}(t)}{\sum_k c_{\alpha k}(t)}.$$

(5)

We assume that resource concentrations are much higher than the species' half-saturating substrate concentrations during the entirety of all temporal niches in which a resource is present. This is a reasonable assumption as long as initial resource concentrations are much larger than their corresponding half-saturating concentrations. Thus, in our model, in each temporal niche, species grow exponentially at constant growth rates given in the previous section as Eq. (1).

During each dilution cycle, the consumer-resource dynamics in our model can be written as:

$$\frac{dN_\alpha}{dt} = g_\alpha N_\alpha$$

(6)

$$\frac{dR_k}{dt} = -\sum_\alpha \left( g_\alpha \cdot \frac{c_{\alpha k} \xi_{\alpha k} x_{\alpha k}}{\sum_k c_{\alpha k} \xi_{\alpha k} x_{\alpha k}} \cdot \frac{N_\alpha}{Y_{\alpha k}} \right).$$

(7)

The coefficient $\frac{c_{\alpha k} \xi_{\alpha k} x_{\alpha k}}{\sum_k c_{\alpha k} \xi_{\alpha k} x_{\alpha k}}$ is the fraction of resource $k$ contributed to the consumption flux of species $\alpha$. The yields $Y_{\alpha k}$ are all set to 1.

The "boom" phase of each cycle ends when all resources are depleted. It is then followed by a "bust" phase during which all microbial abundances are reduced (diluted) by the same factor $D > 1$ until the arrival of the next nutrient bolus, when the boom phase resumes. Our dynamical equations are inspired by the protocol of serial dilution experiments performed in many laboratories to study microbial communities in vitro. A version of this dynamics is also realized in natural microbial ecosystems living in boom-and-bust environments characterized by long intervals between bust phases and large boluses delivered at the beginning of each boom phase.

We are interested in computing the steady state of the boom-and-bust dynamics in which each species grows by exactly the same factor $D$ by which it is subsequently diluted. To compute this steady state dynamics, it is convenient to divide the boom phase of the cycle into a sequence of $n_R$ *temporal niches* where only a particular subset of resources is present [11,33,34]. The total number of possible temporal niches for $n_R$ resources is given by $n_T = 2^{n_R} - 1$. Here we have excluded a temporal niche where all resources are absent and therefore no species can grow. The resources present in the environment disappear one by one, resulting in a particular subset of $n_R$ temporal niches (out of $n_T$ possible ones) realized for each of the $n_R!$ orders of resource depletion. For example, if three resources disappear in the order $2 \rightarrow 3 \rightarrow 1$, the corresponding temporal niches are $111 \rightarrow 101 \rightarrow 100$ (again, we do not show or count the last niche 000, which separates the end of the boom phase and the beginning of the bust phase of the cycle).

Note that we could have also assumed that species can die during the temporal niche where all resources are absent, e.g., due to starvation. Indeed, it is easy to generalize our model to incorporate the possibility of species dying (having a negative growth rate) during any temporal niche. In this situation, competition occurs even in the temporal niche where all resources are absent, adding one more temporal niche to $n_T$. Also, there will be $n_R + 1$ temporal niches between adjacent boom phases.

## Model of lag times

In the version of our model with lag times, we assume that after a resource has been depleted, the species that have been consuming it will enter a lag phase, during which they will reallocate their enzyme pools and will not grow at all. Physiologically, this lag time $\tau$ is the duration required for the enzyme fraction of the new primary resource to increase from its baseline pre-allocation level to its new target level. During this phase, the enzymes necessary to utilize a single resource (or multiple resources in the case of co-utilizers) are synthesized, starting from its pre-allocated level $\phi$ and ending when it reaches the maximum possible fraction $(1 - (n_R - 1))\phi$. This endpoint is determined by the constraint that every other resource in the environment (out of $n_R$ total) must still maintain a minimum allocation of $\phi$. We assumed this process to be autocatalytic, where these metabolic enzymes synthesize precursors (charged tRNA) necessary for their own further production [16], making the synthesis rate proportional to the existing enzyme pool of its own kind: $dE/dt = \tau_0^{-1}E$, where $E$ is the amount of enzyme being synthesized and $\tau_0$ is a time scale. For sequential utilizers, integrating from $t = 0$ to $t = \tau$, we get a lag time $\tau$ as

$$\tau(\text{sequential utilizers}) = \tau_0 \log\left(\frac{1 - (n_R - 1)\phi}{\phi}\right).$$

(8)

For co-utilizers, the log term will include a ratio of the pre- and post-shift allocation fractions, resulting in much shorter lags compared to sequential utilizers.

In Ref. [5], it was proposed and experimentally confirmed that when species switch from glycolytic to gluconeogenic carbon sources, the dynamics of the enzymes necessary to utilize the latter resources are instead given by $dE/dt = \tau_0^{-1}E^2$; this is due to the peculiar biochemistry of these two complementary pathways. In this case we get a different expression for the lag $\tau$ of sequential utilizers as

$$\tau_{\text{glyco}\to\text{gluconeo}} = \tau_0\left(\frac{1}{\phi} - \frac{1}{1 - (n_R - 1)\phi}\right),$$

(9)

along with a corresponding variant for co-utilizers.

## The feasibility of a community assembly

The steady state of a microbial community in a boom-and-bust environment is characterized by a particular depletion order of resources which depends on abundances of both species and resources at the start of the boom cycle in a complex fashion. Since it is not known a priori in which order the resources might be depleted, one should test all $n_R!$ possible depletion orders for feasibility. For a given resource depletion order let $G_{\alpha i}$ be the growth rate of the species $\alpha$ in the temporal niche $i$ ranging from $i = 1$ where all $n_R$ resources are present to $i = n_R$ where only one resource remains. Let $t_i$ be the duration of each of these temporal niches. In the absence of time lags we can calculate the growth ratio of each species during the entire boom phase of the cycle as $\exp(\sum_{i=1}^{n_R} G_{\alpha i}t_i)$. In the steady state, this ratio must be equal to the dilution factor $D$ during the bust phase of the cycle, which leads to the following equations for each of $n_S$ species

$$\sum_{i=1}^{n_R} G_{\alpha i} t_i = \log D. \tag{10}$$

The matrix $G_{\alpha i}$ used in this equation is related to the matrix $g_{\alpha k}$ of growth rates on individual resources, and this relationship depends on the metabolic strategy of the species. If the matrix $G_{\alpha i}$ is invertible, the above equation can be solved, and the solution is biologically and physically feasible provided that all time $t_i > 0$.

In the presence of lags, the steady-state equation can be rewritten as

$$\sum_{i=1}^{n_R} G_{\alpha i} \cdot (t_i - \tilde{\tau}_{\alpha i}) \cdot \Theta(t_i - \tilde{\tau}_{\alpha i}) = \log D, \tag{11}$$

where $\Theta(x)$ is the Heaviside step function, and $\tilde{\tau}_{\alpha i}$ is the modified lag time of species $\alpha$ during temporal niche $i$. In most scenarios, $\tilde{\tau}_{\alpha i} = \tau_{\alpha i}$, where $\tau_{\alpha i}$ is the lag time directly derived from the availability of resources during the temporal niche and the metabolic strategy of the species.

A scenario where $\tilde{\tau}_{\alpha i} \neq \tau_{\alpha i}$ can arise as follows: consider an environment where resources are depleted sequentially in the order $1 \rightarrow 2 \rightarrow 3$. Suppose there is a species $\alpha$ with a resource preference order of $1 \rightarrow 3 \rightarrow 2$. Under short lag phases, species $\alpha$ would switch from resource 1 to resource 3 during the second temporal niche ($t_2$) and would not switch during the third niche ($t_3$). However, if the lag phase is so long that species $\alpha$ remains in the lag phase throughout $t_2$ ($\tau_{\alpha 2} > t_2$) and only switches during $t_3$, we must adjust $\tilde{\tau}_{\alpha 3}$ accordingly. Specifically, $\tilde{\tau}_{\alpha 3}$ is corrected as $\tilde{\tau}_{\alpha 3} = \tau_{\alpha 2} - t_2$.

In contrast, for a species $\beta$ with a preference order of $1 \rightarrow 2 \rightarrow 3$, the lag time during $t_2$ does not contribute to reducing its lag phase in $t_3$, because in $t_3$ it switches to a new resource than in $t_2$. Even if $\tau_{\beta 2} > t_2$, species $\beta$ must undergo a full switching process to transition to resource 3 during $t_3$. Thus, for species $\beta$, the corrected lag time remains $\tilde{\tau}_{\beta 3} = \tau_{\beta 3}$.

With the corrections of $\tilde{\tau}_{\alpha i}$ described above, we used an iterative algorithm to solve for the temporal niches $t_i$ (see code for details). In principle, after performing this algorithm across all $n_R!$ depletion orders for feasibility, one can get more than one feasible solution, corresponding to different ways of assembling a niche-packed microbial community with the same set of species.

## Structural stability with respect to resources

A feasible community with full diversity, given the depletion order of resources, has a unique set of temporal niche durations $t_i > 0$. The next step is to find the range of relative resource concentrations $R_i$ in the nutrient bolus that lead to community assembly. This can be done using the mass conservation rules in the resource-to-biomass conversion. The temporal niche time intervals $t_i$ and lag times $\tilde{\tau}_{\alpha i}$ fully determine the factors $F_{\alpha i} = \exp\left[G_{\alpha i} \cdot \Theta(t_i - \tilde{\tau}_{\alpha i}) \cdot (t_i - \tilde{\tau}_{\alpha i})\right]$, by which individual microbial species grow during that temporal niche. During this time, the biomass of the species $\alpha$ increased from $N_\alpha(0) \prod_{j<i} F_{\alpha j}$ to $N_\alpha(0) F_{\alpha i} \prod_{j<i} F_{\alpha j}$. This increase in biomass is equal to the total resources consumed during this temporal niche. The amount of resource $R_k$ consumed is given by $N_\alpha(0) c_{\alpha k}(\text{during niche i})(F_{\alpha i} - 1) \prod_{j<i} F_{\alpha j}$. This allows one to compute the $M_{\alpha k}$ matrix, which converts species abundances $N_\alpha(0)$ at the start of the boom phase into resource quantities they consumed during the entire boom phase.

$$M_{\alpha k} = \sum_{i=1}^{n_R} \frac{c_{\alpha k}(\text{during niche i}) g_{\alpha k}}{\sum_{k'=1}^{n_R} c_{\alpha k'}(\text{during niche i}) g_{\alpha k'}} (F_{\alpha i} - 1) \prod_{j<i} F_{\alpha j}. \tag{12}$$

Due to mass conservation column sums are given by $\sum_{k=1}^{n_R} M_{\alpha k} = D - 1$. Indeed, in the steady state, the abundance of each species must increase by a factor $D$. This extra biomass given by $N_\alpha(0)(D - 1)$ must be equal to the total quantity $N_\alpha(0) \sum_{k=1}^{n_R} M_{\alpha k}$ of all resources consumed by this species.

The structural stability $S_R$ of a feasible community can be quantified by the fraction of all bolus resource ratios for which the community successfully assembles. It is proportional to the determinant $|\det(M)|$. To properly normalize it, one must first divide the matrix $M$ by $D-1$ so that the sum of the elements in each column is equal to 1. If one randomly chooses $n_R - 1$ ratios between resource concentrations $R_k$ in the nutrient bolus at the beginning of each boom phase of the cycle, the fraction of the volume of the simplex $\sum_k R_k = 1$, $R_k > 0$ that results in community assembly is given by

$$S_R = \frac{|\det(M)|}{(D-1)^{n_R}}$$

(13)

The structural stability defined in this way exponentially scales with the number $n_R - 1$ of independent ratios between resource concentrations. The normalized stability defined by

$$s_R = (S_R)^{1/(n_R - 1)}$$

(14)

corrects for this effect (see Fig E in S1 Text). As can be seen from this study, for $D = 100$, $g_0 = 1$ and $\sigma_g = 0.2$ used in our study, the average value of $s_R$ in communities composed of sequentially utilizing species (both smart and random) is approximately independent of $n_R$. The intuitive interpretation of this quantity is the approximate range of individual nutrient ratios that result in successful community assembly.

In a general case normalized structural stability depends on $n_R$ (the number of resources in the environment), the dilution ratio $D$, the average growth rate $g_0$, and the distribution of growth rates in the $g_{\alpha i}$ matrix.

## Details of numerical simulations

In all of our numerical simulations, we sampled enzyme efficiencies from $\xi_{\alpha i} = \xi_0 + \sigma_\xi Z_{\alpha i}$, where $Z_{\alpha i}$ is a random number drawn from the standard normal distribution. For the figures shown in the main text of the study, we used $\xi_0 = 0.5$ hr$^{-1}$ and $\sigma_\xi = 0.1$ hr$^{-1}$. To prevent negative growth rates and to keep the $\xi_{\alpha i}$ distribution symmetric around its mean, we truncated the normal distribution so that $0 < \xi_{\alpha i} < 1$. The upper bound of the growth rate was set to $g_C = 1$ hr$^{-1}$. All sequential species were set to be top-smart, i.e., with the highest growth rate on their most preferred resource, while their preference order on other resources was randomly generated. The initial lag for each species, where they transition from the dormant state to active growth, was sampled from $\tau_{\text{initial}} \sim U(2, 3)$ hr. The lags caused by enzyme reallocation were generated from eq. 8, where the coefficient $\tau_0$ for each species was sampled from $\tau_0 \sim U(0.2, 0.4)$ hr.

In the simulations described in Fig 2B, for each value of $\phi$, we simulated serial dilution experiments with dilution factor $D = 1000$ and dilution interval $T = 24$ hours. For community assembly, we sampled 200 species pools, each with 50 sequential and 50 co-utilizing species. For head-to-head competition, we sampled 2000 pairs of sequential and co-utilizing species. During community assembly from each species pool, 4 resources added at the beginning of each dilution cycle were sampled uniformly from the simplex $\sum_k R_k = 4$ at the beginning and were fixed for each dilution cycle throughout the assembly process. The duration of each dilution cycle is set to 24 h. We attempted to invade the system multiple times with all species from the pool in a random order until no more successful invasions were possible, i.e., a non-invadable state was reached. All invaders from the pool were introduced with a low abundance of $10^{-8}$, which is much lower than the abundance of any resident species in a steady state community. The elimination bound for a species was also set to $10^{-8}$.

To generate Fig 2C-D, we sampled 100 sequential and 100 co-utilizing species to measure their time-averaged growth rates under monoculture growth. For full diversity communities, we generated 100 purely sequential and 100 purely co-utilizing communities where $n_S = n_R = 4$ species coexist (see S1 Text). We measured the time-averaged growth rate by $\langle g \rangle_T = \log D / T_{\text{dep}}$, where $T_{\text{dep}}$ is the time it takes to deplete the last resource during a dilution cycle at steady state.

In the structural stability simulations in Fig 3C, for each possible composition ($n_S^{\text{sequential}}$=0, 1, 2, 3, 4), we randomly sampled 10000 communities where $n_S = n_R = 4$ species coexist (see Supplementary Text). The $\phi$ for these species was set to 0.036. We then calculated structural stability for these communities (see Supplementary Text).

All simulations were done in Python (see Code Availability Statement).

## Supporting information

**S1 Text. Fig A Relaxing the "top-smart" assumption of sequential species preference order.** Green: Same as the green line in main text Fig 2E, where pool size $N$=100, and all sequential species are top-smart. Red: red lines of different shades (pool size from 100 to 400, with 1:1 fraction of sequential to co-utilizing strategists) correspond to community assembly outcomes, where sequential species have random preference orders. In these simulations, since the probability of being "top-smart" is $1/n_R$ = 1/4, achieving a similar level of sequential species prevalence requires a species pool approximately 4 times larger than in our main simulations (green). **Fig B Ecological effects of metabolic strategies when lags are absent.** (a) Fraction of surviving sequential utilizers in each of the two scenarios: head-to-head (black) and in complex communities (green), plotted as a function of the pre-allocation fraction $\phi$ common to all species in the pool. In the absence of lags, the competitive performance of sequential species is significantly enhanced: compared to the results with lags (Fig 2E), their biomass fraction in head-to-head competitions exhibits as high as ~ 5-fold increase, while the increase in assembled communities is more moderate. The gray horizontal line denotes competitive parity (biomass fraction=0.5). The black vertical line lies at $\phi$ = 0.0615, where sequential species hold a slight advantage in communities, but a slight disadvantage in head-to-head competitions. (b) Distributions of the maximum growth rates sampled from the underlying populations shown in (c); sequential species possess greater extreme values due to their broader distribution tails, enabling them to dominate in complex communities where competition is dictated by extreme value statistics. (c) Population-level growth rate distributions of sequential (blue) and co-utilizing (red) species in the first temporal niche at $\phi$ = 0.0615, highlighting that sequential species exhibit higher variance despite a lower mean.(d)-(e) Time-average growth rates of communities as a function of $\phi$. (d) In monocultures, the sequential utilizers (blue) gain a growth advantage over coutilizers (red) at low $\phi$, which is also reflected in (a). (e) In fully-packed communities, the trend is similar. (f) Distributions of normalized structural stability for niche-packed communities ($n_S = n_R$ species). Apart from the lag time being zero, all other parameters used to generate these communities are identical to the ones in Fig 3C. Increasing the number of sequential utilizers from 0 to 4 systematically increases structural stability. **Fig C Sensitivity of community composition to the initial composition (sequential species proportion) of the species pool** Final biomass fraction of sequential species across varying initial pool compositions (10%–90% of sequential species). Across the regime of small $\phi$ (0 < $\phi$ < 0.05), sequential species exhibit systematic enrichment, often exceeding their initial pool representation and achieving dominance even when starting as a numerical minority. **Fig D Composition of assembled communities stratified by diversity across different values of $\phi$.** (a) Same as Fig 3B. (b)-(f) Stacked bar plot showing how sequential (red) and co-utilizing species (blue) are stratified across communities with different diversities, at specific $\phi$ values marked on (a). **Fig E Schematic of structural stability.** Structural stability quantifies the fraction of supplied resource concentration ratios in which a feasible community can assemble (see Methods). It is defined as the fraction of resource supply ratios that support the coexistence of a given microbial community (green) in all possible resource supply ratios (gray). **Fig F Effects of lags on the ecological advantage of sequential species.** (a) Fraction of sequential utilizers in biomass among survivors with different lags: head-to-head (black) and in complex communities (green), plotted as a function of the pre-allocation parameter $\phi$ common to all species in the pool. Simulations were performed in the same setup as in Fig 2E. Marker shapes indicate the mean coefficient $\bar{\tau}_0$ of proteome reallocation lag. The sequential lag is calculated by $\tau = \tau_0 \log\left(\frac{1-(n_R-1)\phi}{\phi}\right)$ where the coefficient $\tau_0$ is sampled from a uniform distribution $\bar{\tau}_0 \cdot U(0.67, 1.33)$ (Methods). (b) The structural stability of niche-packed communities depends on $\bar{\tau}_0$. Simulations were performed in the

same way as in Fig 3C, where the average logarithm of normalized structural stability (y-axis) is plotted as a function of the composition of niche-packed communities ($n_S = n_R = 4$). Marker shapes represent the value of $\bar{\tau}_0$, and colors reflect the number of sequential/co-utilizing species. When generating these communities $\phi$ was taken at 0.036 (Fig 3B). **Fig G Effects of growth rate distribution on the ecological advantage of sequential species.** (a) Fraction of sequential utilizers in biomass among survivors, in each of the two scenarios: head-to-head (black) and in complex communities (green), plotted as a function of the pre-allocation parameter $\phi$ common to all species in the pool. Simulations were performed in the same setup as in Fig 2E. Marker shapes indicate the width of enzyme efficiency distribution $\sigma_\xi$ of $\xi_{\alpha k}$, which contributes to the growth rates. (b) The structural stability of niche-packed communities depends on $\sigma_\xi$. Simulations were performed in the same way as in Fig 3C, where the average logarithm of normalized structural stability (y-axis) is plotted as a function of the composition of niche-packed communities ($n_S = n_R = 4$). Marker shapes represent the value of $\sigma_\xi$, and colors reflect the number of sequential/co-utilizing species. When generating these communities $\phi$ was taken at 0.036 (Fig 3B). **Fig H Impact of physiological trade-offs on metabolic strategy fitness (a)** Distribution of enzyme efficiencies $\xi_k$ under different sampling schemes: independent sampling (green), $L_1$-norm constraint (yellow), and $L_2$-norm constraint (red). $L_p$-norm constraints are defined by $\sum_k \xi_{\alpha k}^p = \sum_k \xi_0^p$, where $\xi_0 = 0.5$ as specified in Methods. The introduction of $L_p$-norm constraints significantly increases the coefficient of variation ($\sigma_\xi/\mu_\xi$) of efficiencies across resources. The histogram was constructed using $\xi_{\alpha k}$ from 10,000 randomly sampled species for each scheme. **(b)** Biomass fraction of sequential species as a function of the secondary allocation fraction $\phi$. Constraints that impose trade-offs between resources (red and orange curves) shift the dominance of sequential species to larger values of $\phi$, indicating that physiological trade-offs enhance the ecological advantage of the sequential strategy. **Fig I Effect of dilution factor $D$ on the ecological advantage of sequential species.** (a) Fraction of sequential utilizers in biomass among survivors, in each of the two scenarios: head-to-head (black) and in complex communities (green), plotted as a function of the pre-allocation parameter $\phi$ common to all species in the pool. Simulations were performed in the same setup as in Fig 2E. Marker shapes indicate the dilution factor $D$. The growth period is shorter under lower $D$, making the sequential lag's negative effects more pronounced. (b) The structural stability of niche-packed communities depends on $D$. Simulations were performed in the same way as in Fig 3C, where the average logarithm of normalized structural stability (y-axis) is plotted as a function of the composition of niche-packed communities ($n_S = n_R = 4$). Marker shapes represent the value of $D$, and colors reflect the number of sequential/co-utilizing species. When generating these communities $\phi$ was taken at 0.036 (Fig 3B). **Fig J Impact of the dilution factor $D$ on time-averaged growth rates in the absence of initial lag** All simulations shown here were performed by setting the universal initial lag to zero ($\tau_{initial} = 0$). This isolates the specific physiological cost of resource-switching and demonstrates how environmental timescales (governed by $D$) modulate the inherent fitness differences between metabolic strategies. Monocultures (Top Row): Time-averaged growth rates of co-utilizing (red) and sequential (blue) species as a function of the secondary allocation fraction $\phi$ for various dilution factors ($D = 10$, 100, 1000, 10000). In monocultures, increasing $D$ lengthens the growth phase while switching lags ($\tau_{lag}$) remain fixed, thereby amortizing the metabolic switching penalty over a longer cycle and improving the relative fitness of sequential utilizers. Full-diversity pure communities (Bottom Row): Time-averaged growth rates in communities comprising four species of the same strategy ($n_S = n_R = 4$). In these niche-packed contexts, sequential species effectively partition resources into a temporal relay, which minimizes realized lags and renders the steady-state growth rate largely insensitive to $D$ compared to the monoculture case. **Fig K Pool size of assembly affects the ecological advantage of sequential utilizers.** Fraction of sequential utilizers in biomass among survivors (y-axis) are plotted at different pool sizes of the community assembly (represented by different shades of green), as a function of the pre-allocation factor $\phi$. For each value of $\phi$, we randomly generated 200 species pools, and each of them consists of 50% top smart sequential species and 50% co-utilizing species whose growth rates are sampled from the same distribution as in main text (Fig 2D). The gray horizontal line is at $y=0.5$.
(PDF)

## Acknowledgments

We thank Y. Fridman, G. Chure and J. Cremer for valuable discussions.

## Author contributions

**Conceptualization:** Zihan Wang, Akshit Goyal, Sergei Maslov.

**Data curation:** Zihan Wang.

**Formal analysis:** Zihan Wang, Yu Fu.

**Investigation:** Zihan Wang, Akshit Goyal, Sergei Maslov.

**Methodology:** Zihan Wang, Yu Fu, Akshit Goyal, Sergei Maslov.

**Project administration:** Akshit Goyal.

**Resources:** Zihan Wang.

**Software:** Zihan Wang.

**Supervision:** Akshit Goyal, Sergei Maslov.

**Visualization:** Zihan Wang, Akshit Goyal.

**Writing – original draft:** Zihan Wang, Akshit Goyal, Sergei Maslov.

**Writing – review & editing:** Zihan Wang, Akshit Goyal, Sergei Maslov.

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
