## [Decision Letter · Decision Letter 0]

16 Dec 2025

PCOMPBIOL-D-25-02043

Fitness advantage of sequential metabolic strategies emerges from community interactions in strongly fluctuating environments

PLOS Computational Biology

Dear Dr. Goyal,

Thank you for submitting your manuscript to PLOS Computational Biology. After careful consideration, we feel that it has merit but does not fully meet PLOS Computational Biology's publication criteria as it currently stands. Therefore, we invite you to submit a revised version of the manuscript that addresses the points raised during the review process.

We look forward to receiving your revised manuscript.

Kind regards,

Sandro Azaele

Academic Editor

PLOS Computational Biology

Zhaolei Zhang

Section Editor

PLOS Computational Biology

**Additional Editor Comments:**

The reviewers acknowledge that your manuscript presents a thought-provoking theoretical framework for understanding metabolic strategies in microbial communities. While one of them has no major concerns, the other one raises several concerns that require attention before the work can be accepted. The reviewer particularly questions the mechanistic explanation for why sequential utilizers succeed in communities despite resource depletion, and they note that your assumption of independent growth rates on different resources (no physiological trade-offs) may be overly idealized. Additionally, they find the switching between two fitness metrics potentially confusing for readers, and suggest that the "Top-Smart" assumption for sequential utilizers may artificially inflate their advantage. The paper needs more thorough exploration of key parameters (particularly the large dilution factor D), clearer connections to established theoretical frameworks for structural stability, and discussion of how results might generalize beyond the specific serial dilution scenario studied. I think that addressing these points and discussions will significantly strengthen the manuscript.

**Journal Requirements:**

At this stage, the following Authors/Authors require contributions: Yu Fu, Sergei Maslov, Akshit Goyal, and Zihan Wang. Please ensure that the full contributions of each author are acknowledged in the "Add/Edit/Remove Authors" section of our submission form.

5) We have noticed that you have uploaded Supporting Information files, but you have not included a complete list of legends. Please add a full list of legends for your Supporting Information files after the references list.

Potential Copyright Issues:

i) Figures 1, and 2B. Please confirm whether you drew the images / clip-art within the figure panels by hand. If you did not draw the images, please provide (a) a link to the source of the images or icons and their license / terms of use; or (b) written permission from the copyright holder to publish the images or icons under our CC BY 4.0 license. Alternatively, you may replace the images with open source alternatives. See these open source resources you may use to replace images / clip-art:

7) Please amend your detailed Financial Disclosure statement. This is published with the article. It must therefore be completed in full sentences and contain the exact wording you wish to be published.

2) If any authors received a salary from any of your funders, please state which authors and which funders..

8)  Please ensure that the funders and grant numbers match between the Financial Disclosure field and the Funding Information tab in your submission form. Note that the funders must be provided in the same order in both places as well.

9) Thank you for stating 'There are no data associated with this paper. All code is available as a GitHub repository at the following link: https://github.com/maslov-group/Ecol adv diaux' Please note that, though access restrictions are acceptable now, your entire minimal dataset will need to be made freely accessible if your manuscript is accepted for publication. This policy applies to all data except where public deposition would breach compliance with the protocol approved by your research ethics board. If you are unable to adhere to our open data policy, please kindly revise your statement to explain your reasoning and we will seek the editor's input on an exemption.

**Reviewers' comments:**

Reviewer's Responses to Questions

**Comments to the Authors:**

Reviewer #1: In this manuscript, Wang et al. study the competition between microbial species with different metabolic strategies in ecological communities under boom-and-bust cycles. For this, the authors develop a Consumer-Resource model that captures the cyclic arrival of nutrients into the system in a way analogous to serial dilutions experiments. Co-utilizing species divide their enzime budget—and therefore average their growht rate— among the available (or primary) resources, and sequential utilizers use resources one at a time, which can maximize growth rate on their preferred resource at a cost of a potential lag time when switching to another resource.

The paper was a pleasure to read, I found it insightful and well written. I especially appreciate the clear explanation on how smaller enzyme budget fractions allocated to every other primary resource favors sequential consumers in community context, the linear algebra approximation providing an interpretable prediction on the structural stability of niche-packed communities, and the experimentally testable predictions on the prevalence of sequential consumers in high diversity communities—left as a proposal for experimentalists to test in the future. I recommend this manuscript for publication in Plos Computational Biology, with just some minor potential changes. Below are some suggestions and small comments that I have.

MINOR COMMENTS:

I think that the second ‘=‘ sign in Eq. 1 is a typo. I would expect it to be replaced by a product sign ‘·’, or another functional form. Otherwise I don’t understand how both E^r =1 and E^m =1 maximize the growth rate, and an explanation should be given.

The model assumes top-smart sequential species. How would the relaxation of this assumption affect the results? Perhaps it would be interesting to compare with a top-random choice (or any other illustrative choice). Intuitively, this would make it more difficult to find competitive-enough sequential species, increasing the diversity of the species pool required to generate niche-packed communities. But I’d be interested in reading the interpretation—and potentially some analysis— of the authors.

Check the format of all references. For example, there’s a problem with the citation in line [291].

Another typo, this one about Sr in line 372

- Line 107, typo ‘…it takes TO deplete all…’. Check the text for other minor typos like this, e.g. line 122. . In the caption of Fig. S5, some quantity is said to ‘increase by a lot’, which was a bit funny to read (please replace by a more quantitative or rigorous vocabulary). Also in the same caption, ‘y=5’ but ‘y’ has not been defined as biomass fraction. Same caption, the description of panel c precedes the one of b.

- I think that the supplementary figures should be indexed according to their order of citation in the main text. Fig. S5 is the first one to be cited. I think most of the other supplementary figures are cited later in the main text, but I did not check if all of them are cited.

Thanks again for this reading, I found it very interesting.

Reviewer #2: Please see attached

**Have the authors made all data and (if applicable) computational code underlying the findings in their manuscript fully available?**

Reviewer #1: Yes

Reviewer #2: Yes

PLOS authors have the option to publish the peer review history of their article (what does this mean?). If published, this will include your full peer review and any attached files.

Reviewer #1: No

Reviewer #2: **Yes:** Samraat Pawar & Yan Zhu

**Figure resubmission:**
---

## [Decision Letter · Decision Letter 1]

8 Apr 2026

PCOMPBIOL-D-25-02043R1

Fitness advantage of sequential metabolic strategies emerges from community interactions in strongly fluctuating environments

PLOS Computational Biology

Dear Dr. Goyal,

Thank you for submitting your manuscript to PLOS Computational Biology. After careful consideration, we feel that it has merit but does not fully meet PLOS Computational Biology's publication criteria as it currently stands. Therefore, we invite you to submit a revised version of the manuscript that addresses the points raised during the review process.

We look forward to receiving your revised manuscript.

Kind regards,

Sandro Azaele

Academic Editor

PLOS Computational Biology

Zhaolei Zhang

Section Editor

PLOS Computational Biology

**Reviewers' comments:**

Reviewer's Responses to Questions

Reviewer #1: In my opinion, the authors have appropriatly addressed all the questions and suggestions from the referees. I recommend the current version for publication.

Reviewer #2: The revised manuscript presents a clear and conceptually strong contribution, extending consumer–resource models to incorporate dynamic proteome allocation and demonstrating an emergent ecological advantage of sequential metabolic strategies in diverse communities. The authors have addressed the major concerns from the first round effectively, including robustness to key assumptions (e.g. “top-smart” strategies, parameter trade-offs, dilution factor) and improved clarity around the fitness metrics and mechanisms.

We think work is now suitable for publication. However, a small number of minor clarifications and refinements would further strengthen the manuscript and improve accessibility:

* Clarify whether enzymes map one-to-one to substrates or allow multi-substrate use; briefly note implications for lag dynamics.

* Add a short, main-text explanation linking ϕ to microbial physiology and its limitations.

* Briefly signpost early that growth rate is mechanistic, while biomass prevalence is the ecological outcome.

* Include a one-line summary that results are robust (and even strengthened) under constrained efficiencies.

* Clarify more prominently that the mechanism relies on fluctuating environments and is unlikely in steady-state (e.g. chemostats).

* Define “mature communities” at first use.

**Have the authors made all data and (if applicable) computational code underlying the findings in their manuscript fully available?**

Reviewer #1: Yes

Reviewer #2: Yes

PLOS authors have the option to publish the peer review history of their article (what does this mean?). If published, this will include your full peer review and any attached files.

Reviewer #1: No

Reviewer #2: **Yes:** Samraat Pawar & Yan Zhu

**Figure resubmission:**
---

## [Editor Report · Decision Letter 2]

27 Apr 2026

Dear Goyal,

We are pleased to inform you that your manuscript 'Fitness advantage of sequential metabolic strategies emerges from community interactions in strongly fluctuating environments' has been provisionally accepted for publication in PLOS Computational Biology.

Best regards,

Sandro Azaele

Academic Editor

PLOS Computational Biology

Zhaolei Zhang

Section Editor

PLOS Computational Biology

---

## [Editor Report · Acceptance letter]

PCOMPBIOL-D-25-02043R2

Fitness advantage of sequential metabolic strategies emerges from community interactions in strongly fluctuating environments

Dear Dr Goyal,

I am pleased to inform you that your manuscript has been formally accepted for publication in PLOS Computational Biology. Your manuscript is now with our production department and you will be notified of the publication date in due course.

With kind regards,

Judit Kozma
